# Design and Implementation of the MIMO–COOK Scheme Using an Image Sensor for Long-Range Communication

**DOI:** 10.3390/s20082258

**Published:** 2020-04-16

**Authors:** Van Hoa Nguyen, Minh Duc Thieu, Huy Nguyen, Yeong Min Jang

**Affiliations:** Department of Electronics Engineering, Kookmin University, Seoul 02707, Korea; vanhoahd95@gmail.com (V.H.N.); thieuminhduc2011@gmail.com (M.D.T.); ngochuy.hust@gmail.com (H.N.)

**Keywords:** optical camera communication, camera on–off keying, multiple-input multiple-output camera on–off keying

## Abstract

Radio-frequency technologies are widely applied in many fields such as mobile systems, healthcare systems, television and radio broadcasting, and satellite communications. However, one major problem in wireless communication based on radio frequencies is its impact on human health. High frequencies adversely impact human health more than low frequencies if the signal power transgresses the permissible threshold. Therefore, researchers are investigating the use of visible light waves (instead of the radio-frequency band) for data transmission in three major areas: visible light communication, light fidelity, and optical camera communication. In this paper, we propose a scheme that upgrades the camera on–off keying (COOK) scheme by using it with the multiple-input multiple-output (MIMO) scheme; COOK has been recommended by the IEEE 802.15.7-2018 standard. By applying technologies, such as matched filter, region of interest, and MIMO, our proposed scheme promises to improve the performance of the conventional scheme by improving the data rate, communication distance, and bit error rate. By controlling the exposure time, the focal length in a single camera and using channel coding, our proposed scheme can achieve the communication distance of up to 20 m, with a low error rate.

## 1. Introduction

Wireless communication presents advantages over wired communication because it is easier to set up and broadcast data without having to use wires. Wireless communication based on radio frequencies (RFs) has led to the rapid development of mobile networks. However, the radio-frequency resources are being exhausted. To increase the data rate, we need to increase the communication frequencies. Various research groups worldwide have been investigating ways of improving upon the fifth-generation (5G) of digital cellular networks in the frequency band of millimeter waves that increase the data rate (1–10 Gbps). However, it is important to consider that RFs have potentially harmful effects on human health [1]. Therefore, researchers worldwide have attempted to find new technologies to replace RF technologies currently in use. Studies have shown that visible light [2] can be used for data transmission instead of the RF techniques; the optical wireless communication (OWC) technique include three main techniques using LEDs to transmit data [3]: visible light communication (VLC), light fidelity (LiFi), and optical camera communication (OCC). VLC and LiFi both use photodiodes for the reception of signals from LEDs. The difference between VLC and LiFi is not only using visible light spectrum, but infrared spectrum also tested in LiFi [4]. Besides that, LiFi used the bidirectional transmission, with visible light as downlink and the infrared spectrum in uplink when VLC just uses visible light with one-direction. Unlike VLC or Li-Fi technologies that use photodiodes as receivers, the OCC system uses an image sensor as a detector on the receiver side. The performance of the OCC system depends significantly on the camera type [5]. Global shutter and rolling shutter cameras are two types of cameras suitable for the OCC system, and these cameras are available in the commercial market at reasonable prices. For the global shutter camera, the frame rate is the main factor that affects the OCC performance because the frame rate determines whether the sampling rate satisfies the Nyquist sampling requirement. However, in the rolling shutter camera, the sampling rate depends also on the rolling rate of the camera. Their advantages of OWC technique over the RF technique are as follows:Visible light waves do not have any negative influences on the human body [6]. In contrast, RF waves adversely affect human health and reduce system performance because of electromagnetic interference (EMI).The bandwidth of the visible light wave is much greater than that of the RF bandwidth; it is more than 1000 times the RF bandwidth.It is important to consider that when the line-of-sight transmission is gained by the channel model, visible light waves are more safe and efficient.

Given this potential, many companies are investing a substantial amount in funding research on innovations in this technique. OWC technologies were introduced in the IEEE 802.15 standard [7]. The uniqueness of the OWC technique and its complexity protocol are given in [7]. The IEEE 802.15.7m task group (TG 7m) was created to amend the previous standard (i.e., IEEE 802.15.7-2011) [7], which focused only on the VLC mode. In TG 7m, the OWC technique includes the following four features:The updated VLC modes are used in the IEEE 802.157-2011 standard [8].OCC uses cameras or image sensors to receive signals from a light source by using various modulation techniques.Light-emitting diode (LED) identification is a low-speed communication scheme with speeds below 1 Mbps; LEDs are used as photodiodes.Light fidelity (Li-Fi) is a high-speed modulation scheme that uses a high-rate photodiode with data exceeding 1 Mbps in the physical layer.

Currently, the RF technologies cover most areas, such as mobile communication systems, monitoring systems, and radar systems, with typical techniques such as Wi-Fi, ZigBee, and Bluetooth. However, RF frequencies radiate EMI [9]. In [10], the author presented the adverse effects of EMI on human health, especially on brain functions; this fact is now receiving the attention of health researchers worldwide. Recently, the visible light technique has risen as a novel EMI-free substitute for RF technology [11]. In a conventional VLC system, the photodiode receives data according to the fast switching on or switching off of information from LEDs on the transmitter side [12]. In [13], the author implemented an ultra-high-speed pulse density (PD) modulation scheme with extremely effective spectral properties based on the VLC technology. To transmit high-speed data, [14,15] applied multiple-input multiple-output (MIMO) in the VLC/Li-Fi technology with ultra-high-speed multi-channels. When PD schemes are used as detectors, VLC and Li-Fi have a disadvantage that they can only be operated within short distances; this suits the indoor communication system. In an outdoor environment, the effect of the distortion channel impedes the photodiodes from receiving light intensities. To eliminate this, the OCC technology deploys an image sensor as a detector, which can receive data up to 200 m. In [16,17], the researchers assumed that the type of an image sensor (global shutter or rolling shutter) affects the OCC performance. To satisfy Nyquist’s law in the OCC system based on the global shutter camera, the sampling rate needs to focus on the frame rate of the camera. However, with a rolling shutter camera, the sampling rate needs to focus not only on the frame rate of the camera but also on the camera rolling rate. Then, the rolling shutter camera is usually operated for a high data rate system. For long-distance operations and to enhance the signal-to-noise ratio (SNR), the focal length was employed to focus the light beam on the image sensor. Currently, Li-Fi technology is being improved by using a photodiode lens to extend the communication distance, but the maximum distance is only 10 m [18]. 

The OCC system can easily deploy the MIMO technique. It can cope with the big OCC problem of simultaneous connection with many light sources, and it can make it feasible for the camera as well. The MIMO technique is not feasible for the PD-based system when connecting a large number of links. Besides, OCC technology can be operated with many applications based on the image processing algorithm. Region-of-interest (RoI) signaling is an algorithm, which has been applied in OCC [19]. Based on the rolling shutter effect and orthogonal frequency-division multiplexing (OFDM) waveform, the authors of [20] proposed a rolling-OFDM modulation for high-rate communication because when intensity modulation/direct detection (IM/DD) modulation is used, it is difficult to find suitable LED materials comparable with the on-off keying. A two-dimensional OFDM for OCC using the screen method has been discussed in [21]; the researchers achieved a high data rate. However, one disadvantage was the large transmitter size and the short distance, which made it an expensive transmitter.

In [22], we proposed the camera on-off keying (COOK) scheme with a single LED. To decode data, we used the zero-crossing to verify the on/off value in the images. However, certain drawbacks were found. The limited transmission distance and high error rate were compared by using a matched filter to decode the data. In this paper, we propose the MIMO–COOK scheme in which the COOK scheme has been upgraded by using MIMO, region of interest, and matched filter. These technologies are also recommended in [23,24] but we proposed this scheme to upgrade the conventional COOK, which is recommended in IEEE 802.15.7-2018 standard. With this upgrade scheme, the system performance, distance communication and data rate are improved compared with the conventional COOK, which will be illustrated below. 

The remainder of this paper consists of four sections. Section 2 highlights the contributions of the MIMO–COOK scheme. In Section 3, we show the system architecture of the proposed scheme. The implementation results of the proposed scheme are presented in Section 4. Finally, Section 5 concludes this paper.

## 2. Contributions of This Research

In this study, we proposed the MIMO–COOK modulation, which reduces the bit error rate (BER) and improves the data rate as compared with the conventional COOK modulation. Our scheme has the following advantages:Frame rate variation support: frame rate variations have an important effect on the OCC system; packets can be missed when decoding data on the receiver side. Most people believe that the camera frame rate is constantly similar to the camera specifications (e.g., 30 fps or 1000 fps). It is very difficult to synchronize the transmitter (Tx) and the receiver (Rx) depending on the technical parameters of the different cameras. The sequence number (SN) will improve the system performance by checking whether it has a frame rate larger than the packet rate of the transmitter.Detection of missing packets: we used two consecutive images captured by the camera to easily detect each missing packet by comparing two SNs when the length of the SN exceeded a certain value. A comparison of the SN and Ab bit to detect missing packets in the COOK scheme is presented in Section 4.Data merger algorithm: in our experiment, we proposed this algorithm for each data sequence by compacting the packages into it in the exact order. In this scheme, we used the SN to combine the data packets such as the Ab bit.Improve system performance as compared with the conventional COOK: a matched filter is the optimal linear filter for maximizing the SNR. For determining the on/off thresholds in images, the matched filter is superior to the conventional zero-crossing filter. The SNR is maximized; therefore, the communication distance also is increased, and we can achieve the desired BER. This will be shown in Section 3.3.Improved data rate: using multi-LEDs, the data rate will be increased as compared with the conventional COOK. Multi-LEDs are used to transmit data; therefore, RoI signaling is chosen to detect quickly the light sources in images corresponding to each LED.Improve the communication distance: by using the matched filter to detect the start of frame and decode data, the communication distance is improved than the conventional COOK (using Zero Crossing) because of its SNR improvement.

## 3. System Architecture

On off keying (OOK) scheme is known as the simplest form of amplitude-shift keying modulation by using two statuses: ON/OFF to transmit data. In the simplest form of OOK, a binary one is presented by a carrier for a specific duration and a binary zero is presented by the absence of a carrier for the same duration. OOK almost is applied to transfer Morse code via RF waveform or the industrial, scientific, and medical (ISM) band to connect among computers in the computer network because of its simplicity.

In Section 3, we provide the details of the MIMO–COOK scheme. Unlike the COOK mentioned in [22], we used RoI algorithms to detect light sources. To detect preambles and decode, we used a matched filter instead of the zero-crossing filter. Figure 1 illustrates the architecture of MIMO-COOK scheme. The operations of the blocks are presented below.

### 3.1. Line Coding

Line coding is the process of converting digital data to digital signals that convert a sequence of bits into a digital signal. At the sender’s side, the digital data are encoded into a digital signal. At the receiver’s side, the digital data are recreated by decoding the digital signal.

Run-length limited (RLL) coding is a common/useful line-coding technique for a limited bandwidth communication channel. The RLL codes include four parameters: m, n, d, and k. The rate of the code was calculated using the two parameters m and n. However, the two parameters d and k specify the minimum and the maximum number of zeros between consecutive codes. Special RLL codes are used in light communication to provide dc balance because flicker mitigation is required in VLC (except in some OCC modes). In the IEEE 802.15.7-2018 standard, the Manchester codes 4B6B and 8B610 are recommended for deployment in the OCC system [25]. Table 1 presents three line codes, which are recommended in IEEE 802.15.7-2018.

### 3.2. RoI Detection

RoI algorithms are being increasingly used to detect objects because these algorithms provide high accuracy in problem-solving. Almost all RoI algorithms use two approaches: object-based and feature-based [26]. The feature-based approach collects all possible pixels based on the similarity of optical features. Then, these pixels are analyzed to make RoI algorithms. One disadvantage of a feature-based approach is that it is insufficient for identifying the entire target because all pixels do not have a robust feature that is displayed in optical pixels. In contrast, the object-based method can detect RoIs better than the pixel-to-pixel feature-based method because the object-based method uses information about the structures and shapes for detection. The object-based method includes many typical approaches, such as matched filter and template matching [27]. However, the object-based method has certain disadvantages, such as complex calculations and unreliable and low-quality images. In this research, because of using multi-LEDs to transmit data, we need some algorithms to detect and receive data automatically at light sources. RoI emerged as the brightest candidate for it. In this paper, we used the RoI methods to detect the LED objects in each image, which were captured by a camera. It reveals the position of transmitters. Then, the signal from the transmitters can be decoded by receivers.

### 3.3. Match Filter

To highlight the advantages of the matched filter with the COOK scheme, we will compare the matched filter with zero crossing filter (which we used in the conventional COOK scheme). Similar to its name, the zeros crossing filter distinguishes the ON/OFF levels in the receiver side based on the zero levels as the threshold level. This method will have a good performance if the SNR of the system is high, but with the SNR lower, the distinction between ON/OFF statuses is unclear as shown in Figure 2. Then if we depend on the zero levels to define the threshold level, the performance of the system is reduced. From that, we can conclude that the zero crossing filter becomes ineffective with the low signal to noise system. In Section 4, we demonstrate the relationship of SNR, exposure time of camera, and communication distance. With the same exposure time, the SNR will be reduced if the communication distance is high. From that, we need another technique instead of the zero crossing filter to improve the communication distance and the system performance.

The matched filter is a filter technology achieved by comparing a template signal with the real signal to determine the template signal in the real signal. The matched filter, which is one of the linear filter technologies, optimizes the SNR in the appearance of additive random noise. Matched filters are widely used in almost all wireless communication systems, such as mobile communications and radar systems to maximize the SNR of the system; these filters increase the system performance. In the IEEE 802.15.7-2018 standard [25], the matched filter is also recommended for use in the OCC system. The performances of the matched filter and zero-crossing filter in the COOK scheme have been compared in Section 4. In this paper, we used the matched filter for two purposes: detect preamble and decode data.

To detect preamble positions, the received signal is multiplied with the known preamble signal via the convolution algorithm as Equation (1). The preamble positions in each image are the peak of the resulting signal as Figure 3. From that, we can explore payload positions.
(1)$$\left(f*g\right)\left(t\right)≜\overset{+\infty }{\underset{-\infty }{\int }}f\left(\tau \right)g\left(t-\tau \right)d\tau$$

After the detect preamble, to decode data, we also used the matched filter to decode data to improve the system performance. By create known patterns as Figure 4, the received signal is multiplied with the known preamble signal via the convolution algorithm. From convolution results, we can know that: which patterns are the most like the received signal? From that, it is easy to verify the value of signals (0 or 1). Same with the Manchester code, we can create 16 patterns of 4B6B code to decode data.

## 4. Implementation

### 4.1. Experimental SNR Measurement

The SNR was measured to demonstrate the relationship between the communication distance, SNR, and exposure times. We can adjust the exposure time to achieve the desirable BER value and communication distance for the OCC system. In the experiment, we conducted many measurements for the distance between 5–20 m to calculate the SNR. As shown in Figure 5, we used LED (12-V, 3-W, 2 × 2 cm^2^ rectangle) as a transmitter and a rolling shutter camera (Point Gray) as a receiver. We also set up the exposure times of the camera in the range of 50–500 μs with appropriate distance.

The SNR was measured at a distance of 5, 10, 15, and 20 m (Figure 6, Figure 7, Figure 8 and Figure 9). We can use two status of LED to calculate SNR. The first status of LED is ON, the SNR can be considered as the signal power of the LED and is displayed as a red line at a different distance. The second status of the LED is OFF, we used a blue line to display the SNR which can be awarded as background noise at this status. The SNR (dB) can be figured out based on the experiment by using the following equation:(2)$$\begin{array}{l} SNR_{dB}=20\log\left(\frac{\sqrt{\frac{1}{n}{\sum_{i=0}^{n-1}\left|A_{i}\right|}^{2}}}{\sqrt{\frac{1}{n}{\sum_{i=0}^{n-1}\left|B_{{}_{i}}\right|}^{2}}}\right) \end{array}$$
here, A is represented for the signal power of the LED (i.e., pixel amplitudes which are measured when the LED turned on); B is represented for the background noise (i.e., pixel amplitude which is measured when the LED is turned off); and the number of measured samples is represented by n.

The received pixel amplitude values will be large at a short distance and small at a long distance. The shuttle speed and exposure time also affect the SNR calculation. The relationship between the SNR value and pixel intensity with a different exposure time of 50, 150, 300, and 50 μs is illustrated clearly in Figure 7. The most important factor that affects the SNR is exposure time. In this scenario, the operation of the image sensor is similar to a low-pass filter. Since the high-tone signal will be smoothened when the exposure time is greater. Moreover, the communication bandwidth will go down as well which significantly affects to decrease in the total noise power spread over the bandwidth when the exposure time increases. Long exposure time also leads to the rise in the SNR. Besides that, the relationship between the SNR and the communication distance is illustrated in Figure 10. In the longer distance, the SNR value is reduced because of the power of light sources received in image sensors reduces. As we mentioned before, the matched filter is one of the linear filters to optimize the SNR of the system, then it also improves the communication distance. The comparison of COOK performance between zero-crossing and matched filter is illustrated in Section 4.4.

### 4.2. Simulation BER of COOK Scheme

In the receiver side of the communication system, the electrical amplitude of the received signal is illustrated as:(3)$$r(t)=I(t)+\sum_{i=-\infty }^{+\infty }I(t)a_{i}g(t-iT_{symbol})+n(t)$$
where $a_{i}$ is the level of the $i^{th}$ OOK symbol and $a_{i}$∈ {0,1}; we assumed that the probabilities of bit 0 and 1 are $P_{0}$ and $P_{1}$, g(t) presents as the rectangular function, and $T_{symbol}$ is the symbol duration. Because of the only additive white Gaussian noise (AWGN) in channel model, the BER can be presented [26] in Equation (4).
(4)$$P_{e}=\frac{1}{2}erfc(\sqrt{\frac{E_{b}}{2\sigma_{n}^{2}}})$$

Because of the only AWGN channel, the received signal in the receiver side for bits “0” and “1” is shown as follows:(5)$$r(t)=\left\{\begin{array}{l} n(t),a_{i}=0\\ 2I(t)+n(t),a_{i}=1 \end{array}\right.$$

Figure 11 shows the estimated curve of the COOK scheme versus the SNR with the matched filter and zero-crossing filter compared with the theoretical bit error probability of OOK modulation based on the additive white Gaussian noise channel. The figure demonstrates that the use of a matched filter to decode the COOK scheme is better than the zero-crossing filter. It illustrates that the performance of our proposed scheme is better than that of the COOK proposed in [22]. From the SNR measurements of Figure 10 and Figure 11, we can gain a BER of $10^{-4}$ at 20 m (SNR of 12.5 dB, exposure time of 50 μs) using the matched filter. However, with the zero-crossing filter, we need an SNR of 15 dB corresponding to the distance of 15 m and an exposure time of 50 μs. The exposure time has a significant role in the SNR measurement because the SNR improves with an increase in the exposure time. However, a long exposure will reduce the communication bandwidth. To achieve the best performance, it is important to set the exposure time carefully depending on the application. Besides that, the COOK using matched filter improved the performance significantly, which is almost the same with the theoretical bit error probability of OOK modulation in $E_{b}/N_{0}$ values of 0–8 dB. This facility to improve the SNR values of the system without changing the exposure time, from that increase the distance communication without reducing the system performance.

### 4.3. Proposed Scheme

Figure 12 shows the format of the proposed scheme that we will discuss in this subsection. In this scheme, every packet contains many sub-packets to support the compatibility of the frame rate variations. The serial number of the sub-packet was created as an SN. The sub-packet in the same packet had the same data payload. In practice, depending on the combination of the packet rate of the transmitter and the frame rate of the camera, the SN was divided and controlled. When the frame rate was smaller than the packet rate, undersampling occurred. In contrast, when the frame rate was larger than the packet rate, oversampling occurred. The packet rate is the number of packets (containing different payloads) being constantly transmitted in one time period (e.g., 20 packets/s). The proposed data frame of our scheme contained data packet frames that contained smaller data sub-packets (DSs); every DS holds the payload data and an SN. The SN carries the sequence information of a data packet and the identity of the payload. Payload identification was useful in practice because it helped the receiver identify the arrival state of a new payload in the oversampling case and detect lost payloads in the undersampling case. An example of SN in practical studies [22] is where the 1-bit Ab is used in the oversampling case, and 2-bit Ab is used for the undersampling case. A disadvantage of this method is the confusion of the receiver in the decoding process and the limited missing payload that it can detect. The proposed scheme can overcome this disadvantage by using n bits of the SN. The length of the SN can be changed based on the practical situation and the system parameters. Depending on the channel environment, the SN length is chosen suitably. In harsh environment or a bad camera quality, the missed packet is large, then we need the SN of large length. However, with a good camera quality, we can reduce the SN length to improve the system performance. In our paper, we used the SN length of 4 bits to implement the MIMO-COOK scheme.

Same with the COOK scheme, Figure 12 shows how to use SN with oversampling and undersampling cases. Besides that, to identify the order of LEDs in MIMO–COOK, we proposed LED_ID, which helped combining each separated packet in each LED in the order of the images.

#### 4.3.1. Oversampling

When the frame rate of a rolling shutter camera is at least two times larger than the packet rate of light sources, the data packet is sampled multiple times causing the oversampling effect. It occurs because of the frame rate variation effect, then we need to use the repetition code to transmit data. When the packet is sampled more than once, errors of packet merger are created at the receiver side. The SN is added to the DS to deal with this problem because it improves the receiver’s ability to decrease the frame rate variation effect. Redundant data were removed from the receiver when the same SN value is recognized in the DS of different packets. The receiver side can exclude consecutive packets with the same SN value and select packets with consecutive SN values (n − 1, n, n + 1) for the merger, as illustrated in Figure 13.

In our proposed algorithms, the data sub-packets are added to the data packet; this is repeated N times to reduce the loss rate of the packets while the camera cannot capture the time between the sub-packets. To enhance the reliability of the OCC system, we need to prevent the missing of packets when there is a change in the frame rate of the camera. For this, the calculation of the N value must conform to the following equation:(6)$$N\geq \frac{{\left(T_{cam}\right)}_{\max}}{DS\_length}$$
where $T_{cam}$ is the inter-frame interval between two consecutive images, and DS_length is the data packet interval.

#### 4.3.2. Undersampling

When the frame rate of the camera decreases below the packet rate of the transmitter, undersampling occurs. The payload will be lost in undersampling unlike in the case for oversampling. Figure 14 shows the scenario in which the missing payload is created and detected using the SN. In this case, if the SN is long enough, the receiver can detect the missing payload. If one payload is missing, the error can be detected by comparing the SN of the two adjacent DSs which received on the receiver side. The number of SNs in different states depend on the length of SN part in each sub-packet. For example, four missing payloads of the transmitted packets can be detected if we use two bits for the SN length. When one error is detected, the error correction becomes easier. If two consecutive sub-packets have two non-consecutive sequence number values (n − 1 and n + 1), missings occur as shown in Figure 14. In our implementation below, we use the SN length of 4 bits then 16 missing payloads can be detected if the undersampling case occurs.

#### 4.3.3. Maximum Length of Sub-Packet

The time interval for scanning one of the pixels in each image is called the rolling rate of the rolling shutter camera [28]. The number of pixels considered as a bit are counted as Equation (7):(7)$$N_{pixel/bit}=\frac{T_{clock}}{T_{rolling}}$$

According to Nyquist’s theorem, certain new features are available in the rolling shutter camera. The condition for the optical clock rate of LEDs based on the shutter speed of rolling shutter camera is $T_{clock}\geq 2T_{rolling}.$
(8)$$N_{symbol}\leq \frac{LED\_size}{N_{pixel/bit}}$$

### 4.4. Implementation Results

Our experiments were performed ten times with the MIMO–COOK scheme, and some rolling shutter cameras were used to verify the frame variation effect. The asynchronous process includes asynchronous decoding, merging data technique, and the detection of missing parts. The SN length also is chosen carefully for the overall process. For the Point Gray rolling shutter camera, the frame rate variation was between 20 fps and 60 fps. Figure 15 shows the setup of implementation, and Figure 16 shows the practical MIMO–COOK waveform. The experimental results are shown in Figure 16, and the results and parameters are shown in Table 2.

Figure 17 depicts the implemented results; the experiments were performed ten times with different communication distances at the same exposure time of 100 us for comparing the COOK with our proposed scheme. Figure 17 shows that with the same communication distance and channel environment, the matched filter significantly improved the system performance more than the zero-crossing filter. From Figure 10, we can modify the pixel SNR by setting an exposure time to improve the system performance at a long communication distance; however, the high exposure time will increase the probability that the fuzzy states of LEDs lead to a reduction in the communication bandwidth. From that, we can conclude that: by using the matched filter, the communication distance can improve without reducing the system performance. Besides this, to improve the system performance at longer communication distances, channel coding is recommended to mitigate the high ratio of fuzzy states.

Table 2 shows the experimental results obtained by conducting experiments several times using different optical clock rates. The table shows that the data rate can increase by increasing the data sub-packet length and the optical clock rate; however, the clock rate and the data sub-packet length must suit the rolling shutter camera parameters, namely, the rolling rate of the rolling shutter camera and the size of images, as shown in Equations (7) and (8). We implement this scheme with two LEDs at the communication distance of 2 m, which can be visually assessed by the Appendix A.

## 5. Conclusions

In this paper, we proposed the MIMO–COOK scheme, which was an upgrade of the conventional COOK scheme. The MIMO–COOK scheme uses a matched filter instead of a zero-crossing filter; this significantly improved the performance of the OCC system. The proposed frame structure using SN can support the frame rate variation effect and recover data in images. Furthermore, we deployed the SNR measurements with different exposure times and distances to verify the relationship between the exposure time, SNR, and communication distance. Finally, we measured the BER of this scheme with different communication distances and compared these with the conventional COOK scheme.

By inserting the sequence number on each sub-packet, we can connect the sub-packet from other images to become the huge packet, which is the desired transmitted signal. By using the SN length which is long enough, the lost packets can be detected based on the non-sequentials of SNs leading to improved BER performance. However, the trade-off between the BER performance and the data rate needs to be considered carefully because the SN is a redundant component that does not carry data.

Besides that, by using the matched filter to detect preamble and decode data, the system performance is improved compared with the conventional COOK scheme using zero-crossing. Because of the increase in SNR value, the matched filter also increases the communication distance without changing the system performance compared with the conventional COOK scheme.

## Figures and Tables

**Figure 1 sensors-20-02258-f001:**
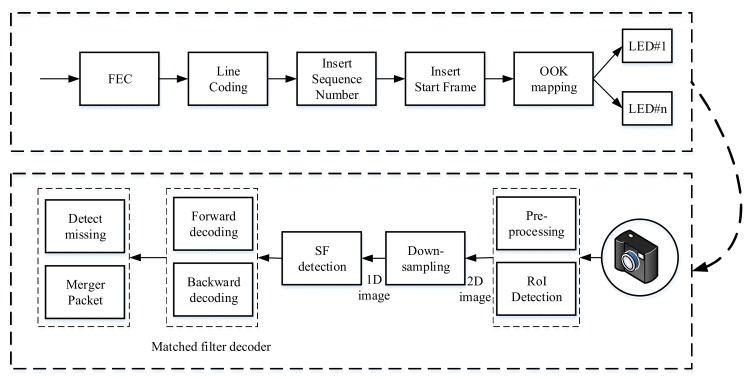
Reference architecture of the multiple-input multiple-output–camera on–off keying (MIMO–COOK) scheme.

**Figure 2 sensors-20-02258-f002:**
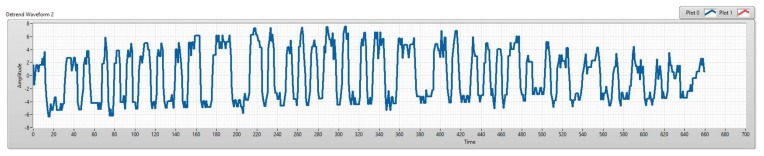
An experimental result of COOK signals within a rolling image with low signal-to-noise ratio (SNR).

**Figure 3 sensors-20-02258-f003:**
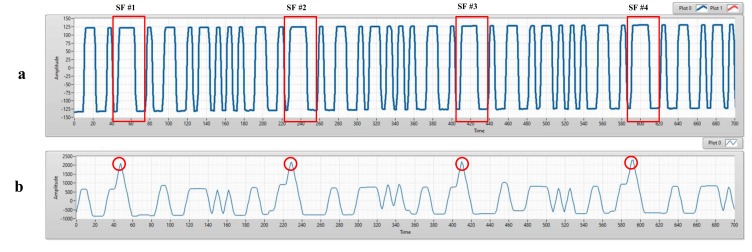
(**a**) An experimental result of COOK within a rolling image. (**b**) The results of preamble position detection based on matched filter.

**Figure 4 sensors-20-02258-f004:**
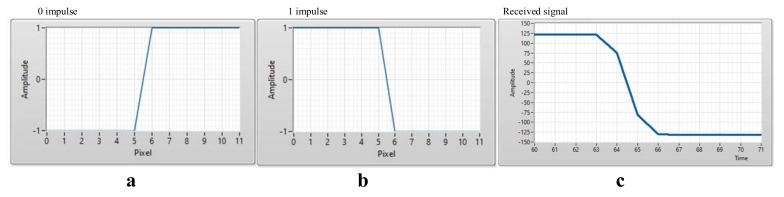
Manchester code signal patterns and COOK received signal. (**a**) 0 impulse, (**b**) 1 impulse, (**c**) COOK received signal.

**Figure 5 sensors-20-02258-f005:**
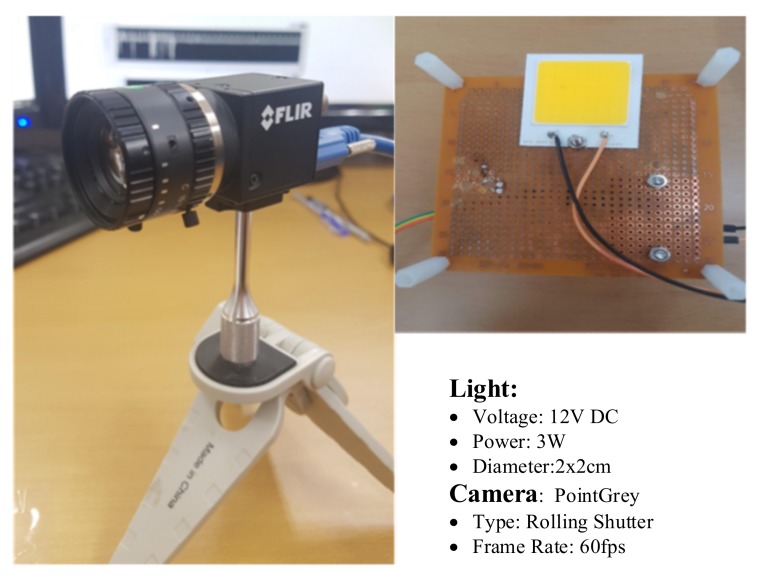
Device for SNR measurement.

**Figure 6 sensors-20-02258-f006:**
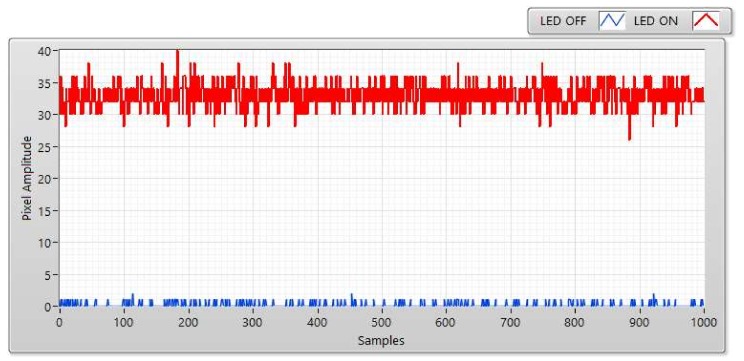
SNR measurement with Point Gray rolling shutter camera at communication distance of 5 m and exposure time of 50 μs.

**Figure 7 sensors-20-02258-f007:**
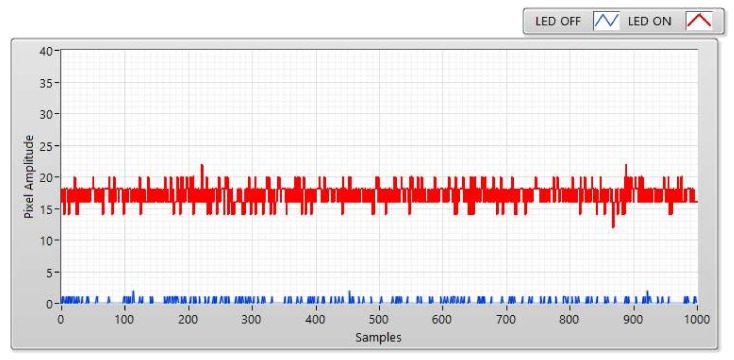
SNR measurement with Point Gray rolling shutter camera at communication distance of 10 m and exposure time of 50 μs.

**Figure 8 sensors-20-02258-f008:**
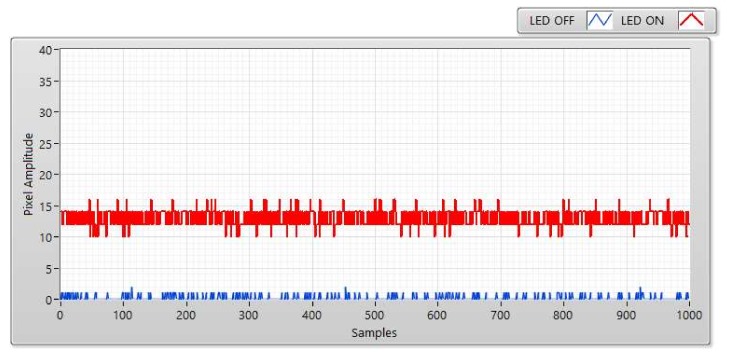
SNR measurement with Point Gray rolling shutter camera at communication distance of 15 m and exposure time of 50 μs.

**Figure 9 sensors-20-02258-f009:**
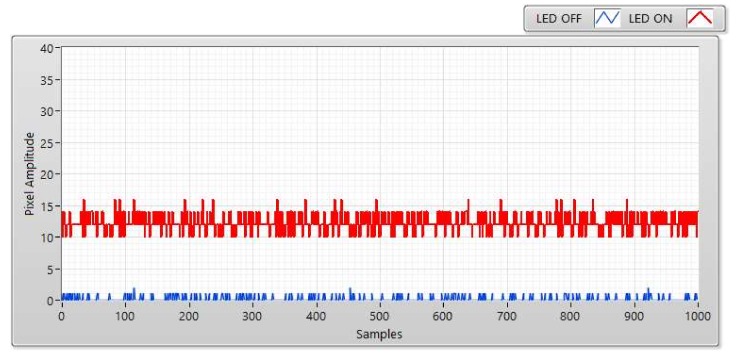
SNR measurement with Point Gray rolling shutter camera at communication distance of 20 m and exposure time of 50 μs.

**Figure 10 sensors-20-02258-f010:**
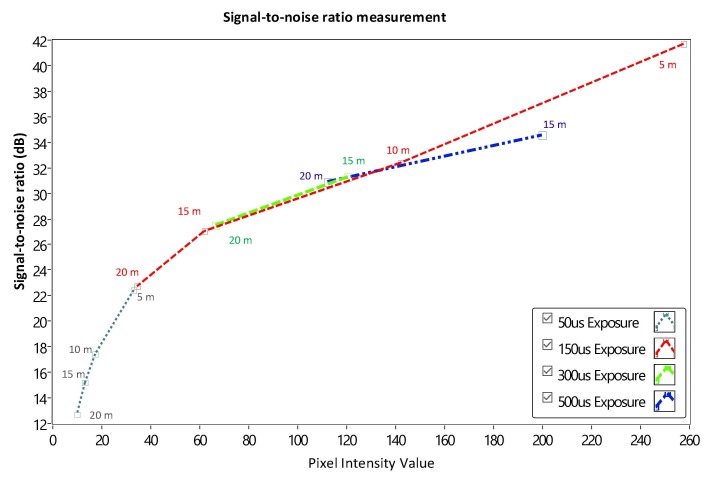
SNR measurement.

**Figure 11 sensors-20-02258-f011:**
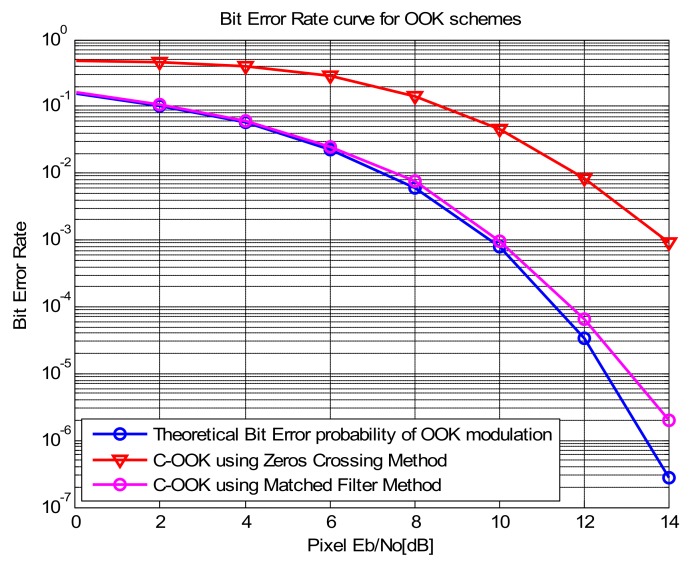
Comparison of bit error rate (BER) of the C-OOK scheme using a matched filter and zero-crossing techniques with the theoretical Bit Error probability of OOK modulation.

**Figure 12 sensors-20-02258-f012:**
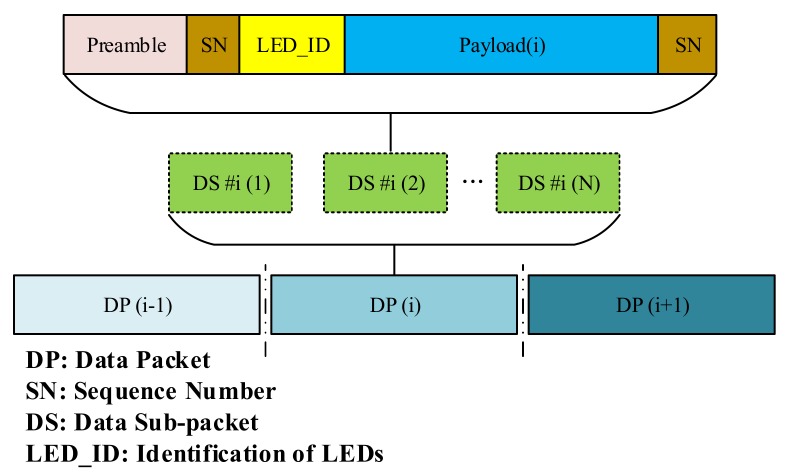
MIMO–COOK data packet structure.

**Figure 13 sensors-20-02258-f013:**
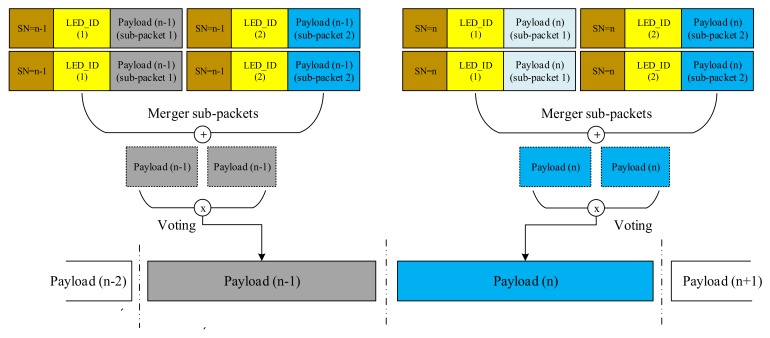
Merging procedure of MIMO COOK scheme in the time domain.

**Figure 14 sensors-20-02258-f014:**
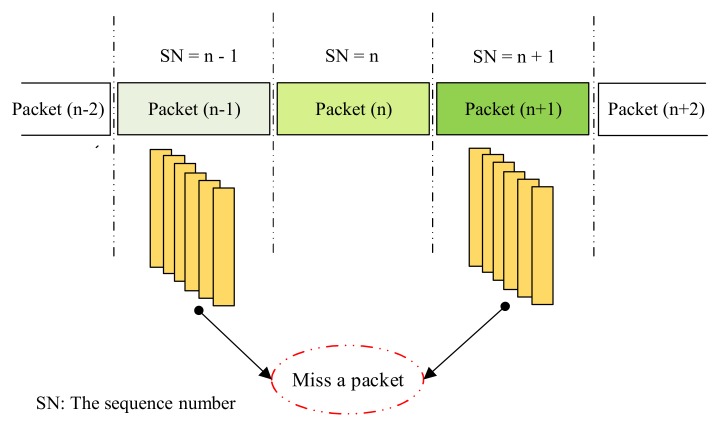
Error detection in grouping image during undersampling case.

**Figure 15 sensors-20-02258-f015:**
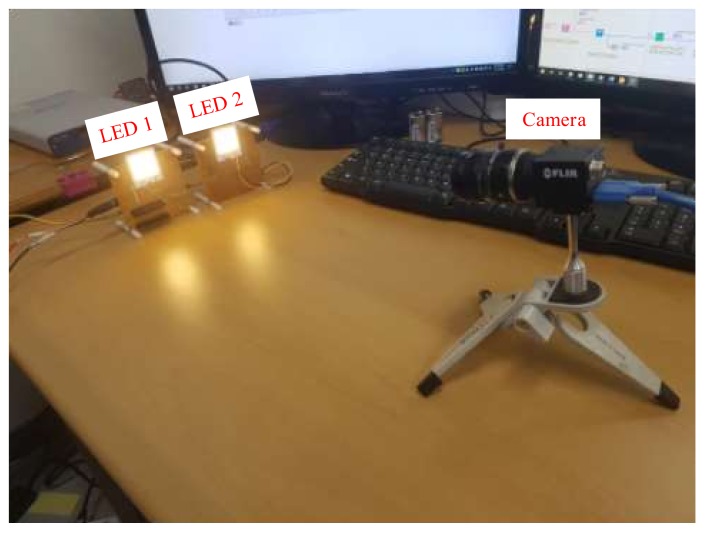
Setup for the MIMO–COOK implementation.

**Figure 16 sensors-20-02258-f016:**
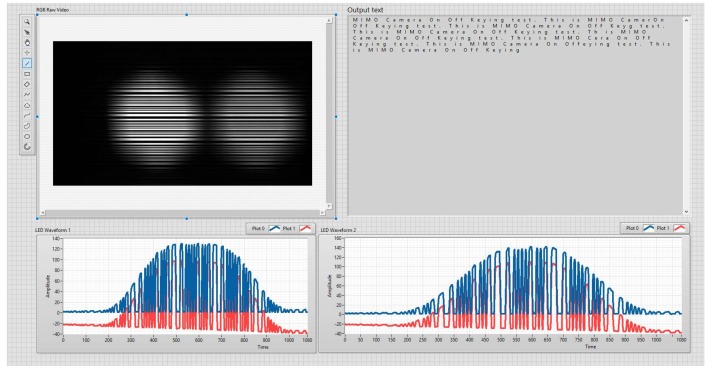
Rx interface.

**Figure 17 sensors-20-02258-f017:**
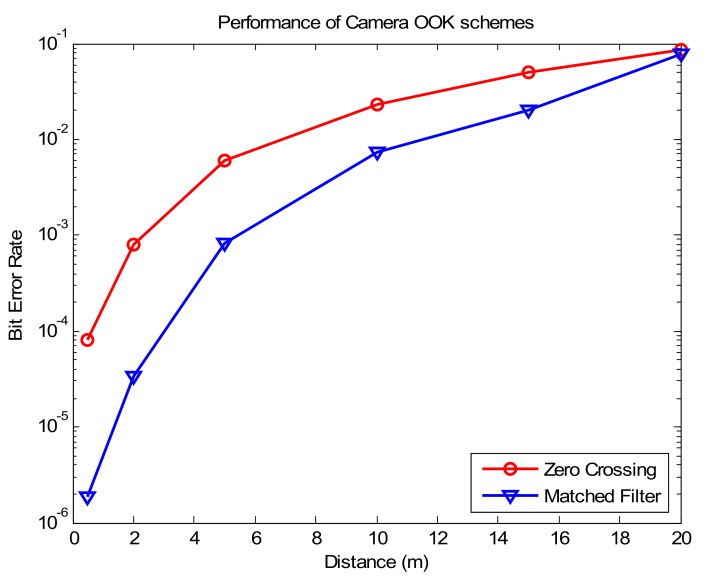
BER of the COOK system with an exposure time of 100 us.

**Table 1 sensors-20-02258-t001:** Coding scheme and proposed preamble.

RLL Code	Proposed Preamble (SF)
Manchester	011100
4B6B	0011111000
8B10B	0000111111111100000

**Table 2 sensors-20-02258-t002:** MIMO–COOK parameters.

**Transmitter Side**				
**Clock rate**	8 kHz		10 kHz	
**RLL**	Manchester	4B6B	Manchester	4B6B
FEC	RS (15,11)			
LED type	12 V, 3 W			
Number of LEDs	2			
Packet rate	20 packet/s			
**Receiver side**				
Camera	Point Gray rolling shutter camera			
Camera frame rate	60 fps			
**Data rate**				
Uncode bit rate	1.2 kbps	1.8 kbps	1.5 kbps	2.25 kbps
Code bit rate	0.88 kbps	1.32 kbps	1.1 kbps	1.65 kbps

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
