# Peer review of "Design and Implementation of the MIMO–COOK Scheme Using an Image Sensor for Long-Range Communication"

_sensors, 2020, doi:10.3390/s20082258_

Round 1

Reviewer 1 Report

  • What are the major advantages of the proposed solution in comparison with the existing approaches?
  • Apart from the engineering solution, what research elements are included that can be further enhanced to have better results?
  • The mathematical principles behind the setup and functioning of the proposed model need to be discussed which can give a clear understanding of the observations.

Author Response

Manuscript ID: sensors-736187

Article Title: “Design and Implementation of the MIMO–COOK Scheme Using an Image Sensor for Long-Range Communication”

To: Sensors Journal Editor and Reviewers

Re: Response to reviewers

Dear Editor and Reviewers.

Thank you for allowing a resubmission of our manuscript, with an opportunity to address the reviewers’ comments.

In this resubmission, we have tried our best to respond to all the concerns of three reviewers with our careful and detailed responses. All of the remaining concerns have been clarified in this resubmission.

In this manuscript, we have tried our best to present the technical feasibility of our proposed system by numerous implementation results. We hope that our revisions and explanations will address three reviewer concerns and requirements.

We are uploading an updated manuscript with red color text indicating changes.

Best regards,

Van Hoa Nguyen

Minh Duc Thieu

Huy Nguyen

Yeong Min Jang

Reviewer 2 Report

See attachment

Author Response

(The authors gave the same response as above.)

Reviewer 3 Report

In this paper, the authors investigate a MIMO-COOK scheme for OCC systems and experimentally demonstrate a kbps transmission over 20m distance. Overall, this paper presents some intresting techniques and reports promising experimental result, but its novelty and conctributions are not clear. As a result, this reviewer believes that proper revisions are required before a final recommendation can be made. 

The detailed comments are given as follows:

1. This work involves several different techniques such as MIMO, camera OOK, matched filter, region of interest, etc. However, nearly all these techniques can be found in the existing literature. Therefore, the authors should clearly highlight the true novelty and unique contributions of this current work in comparion to their previous works and other published ones.

2. The authors divide the communicaiton approach using visible light waves into three categories, including visible light communication, optical wireless communication, and optical camera communication. In this reviewer's opinion, this kind of categorization is not appropriate, since, for example, visible light communication can also be considered as optical wireless communication which uses visible light and optical camera communication is also one kind of visible light communication which adopts camera as receiver.

3. The English writing needs to be substantially improved. One example can be seen in page 2, line 81.

4. More experimental results are expected to show the performance compariosn between the proposed scheme and the existing ones.

5. Some recent relavant works are recommended to be included in the introduction of OCC. i.e., 10.1109/MCOM.2018.1601017, 10.1109/TMC.2017.2694834.

Author Response

(The authors gave the same response as above.)

Round 2

Reviewer 2 Report

The contribution of the revised version has been improved compared to the previous version. The authors have added some useful materials and results with better explanations. However, there are still some minor corrections need to be considered. The SN length is still not given in the revised manuscript while Fig.16 is still not readable. The abbreviation of OOK modulation should be defined. It must put a dot (.) after at the end of the figures and tables title while space is needed before [3] in line 40. I do recommend the authors to put the figures after they mention inside the text such as Figures 6-10.

Reviewer 3 Report

I only have one final comment regarding the categorization of OWC. In the revised papre, the authos have revised the categorization and claimed that OWC have three techniques, i.e., VLC, LiFi and OCC. Actually, this kind of categorization is also questionable. For example, VLC can only be considered as a part of LiFi, since LiFi is generally bidirectional which adopts VLC as downlink and infrared communication as uplink. So, it is not appropariate to place VLC and LiFi parallelly. The authors are required to make a more in-depth litereature reivew regarding the categorization of OWC. One potential suggstion from this reviewer is to see from the wavelength/frequency point of view.
